# Saffron: The Golden Spice with Therapeutic Properties on Digestive Diseases

**DOI:** 10.3390/nu11050943

**Published:** 2019-04-26

**Authors:** Hassan Ashktorab, Akbar Soleimani, Gulshan Singh, Amr Amin, Solmaz Tabtabaei, Giovanni Latella, Ulrike Stein, Shahin Akhondzadeh, Naimesh Solanki, Marjorie C. Gondré-Lewis, Aida Habtezion, Hassan Brim

**Affiliations:** 1Department of Medicine, Department of Pathology and Cancer Center, Howard University College of Medicine, Washington, DC 20059, USA; akbar.soleimani@Howard.edu (A.S.); hbrim@howard.edu (H.B.); 2Division of Gastroenterology and Hepatology, School of Medicine, Stanford University, Stanford, CA 94305, USA; gsingh10@stanford.edu (G.S.); aidah@standford.edu (A.H.); 3Biology Department, UAE University, Al Ain 15551, UAE; a.amin@uaeu.ac.ae; 4Department of Chemical Engineering; Howard University, Washington, DC 20059, USA; solmaz.tabtabaei@Howard.edu; 5Gastroenterology, Hepatology and Nutrition division, Department of Life, Health and Environmental Sciences, University of L’Aquila, 67100 L’Aquila, Italy; giolatel@tin.it; 6Experimental and Clinical Research Center, Charité-Universitätsmedizin Berlin and Max-Delbrück-Center for Molecular Medicine, 13125 Berlin, Germany; ustein@mdc-berlin.de; 7German Cancer Consortium (DKTK), 69120 Heidelberg, Germany; 8Psychiatric Research Center, Roozbeh Hospital, Tehran University Medical Sciences, Tehran 14167-53955, Iran; sakhond@yahoo.com; 9Neuropsychopharmacology Laboratory, Department of Anatomy, Howard University College of Medicine, Washington, DC 20059, USA; naimesh.solanki@gmail.com (N.S.); mgondre-lewis@Howard.edu (M.C.G.-L.)

**Keywords:** saffron, gastrointestinal diseases, microbiome, colon, stomach, liver, pancreas, inflammation, gastritis, hepatitis: pancreatitis, colitis, cancer

## Abstract

Saffron is a natural compound that has been used for centuries in many parts of the world as a food colorant and additive. It was shown to have the ability to mitigate various disorders through its known anti-inflammatory and anti-oxidant properties. Several studies have shown the effectiveness of saffron in the treatment of various chronic diseases like inflammatory bowel diseases, Alzheimer’s, rheumatoid arthritis as well as common malignancies of the colon, stomach, lung, breast, and skin. Modern day drugs generally have unwanted side effects, which led to the current trend to use naturally occurring products with therapeutic properties. In the present review, the objective is to systematically analyze the wealth of information regarding the potential mechanisms of action and the medical use of saffron, the “golden spice”, especially in digestive diseases. We summarized saffron influence on microbiome, molecular pathways, and inflammation in gastric, colon, liver cancers, and associated inflammations.

## 1. Introduction

*Crocus sativus*, a plant of the iris family (Iridaceae), is largely cultivated in countries around the Mediterranean Sea and parts of Asia, namely in Iran, India, Italy, Spain, Greece, and Morocco. Its flower contains various chemical compounds [1]. Stigmas of the flower (saffron) contain bitter principles (e.g., picrocrocin), volatile agents (e.g., safranal), coloring substances (e.g., crocetin and its glycoside crocin) anthocyanin, carotene, and lycopene [2]. These constituents have been described as having various pharmacological effects such as anti-depressant and anti-cancer properties [3]. Crocin is a main constituent in the hydro-ethanolic saffron extract [4] that has effects such as anti-convulsant, anti-depressant, anti-inflammatory, anti-tumor, radical scavenger capacity, learning, and memory-improving properties [5].

The role of medicinal plants in the treatment of various disorders such as gastrointestinal (GI) diseases and psychological disorders has become well established over the past decade, with known phytotherapeutic preparations (e.g., St. John’s wort (SJW) and *Piper methysticum* a.k.a. Kava). In traditional Persian medicine, saffron is used against depression, with four randomized controlled human clinical trials currently underway supporting this use [6]. However, saffron is not recommended for use by pregnant women where it might interfere with embryo development [7]. Medicinal plants are used by about 80% of the world population, primarily in the developing world. They stood the tests of safety, efficacy, acceptability, and lower side effects. Their chemical constituents are believed to have better compatibility with the human body [8]. In this review, we focused on the potential mechanisms of action of saffron and its beneficial effects on several digestive tract diseases.

Message of the Manuscript: The wealth of information regarding the potential mechanisms of action and the medical use of natural herb, saffron, particularly in digestive diseases. Saffron influence on microbiome, molecular pathways, and inflammation in gastric, colon, liver cancers, and associated inflammations.

## 2. Methods

To capture the known and potential medical benefits of saffron, we performed a comprehensive review of the literature on saffron and its applications in the field of GI diseases. Literature search was performed using PubMed, Medline, and Google Scholar to search for publications related to saffron. The following keywords were used “saffron”, “crocin”, “crocetin”, “safranal”, “colon”, “gastric”, “colitis”, “Irritable Bowel Syndrome”, “Inflammatory Bowel Disease”, “chemoprophylaxis”, “cancer chemoprevention”, “Apoptosis”, “cell cycle”, “colorectal cancer cell lines”, “hepatocellular carcinoma”, “metastasis”, and “anti-inflammatory”. All collected papers were reviewed by authors. The objective of this review was to present the published gastrointestinal data on saffron to date with the intent to increase our understanding of barriers that are withholding the medical community from its clinical use. 

## 3. Saffron Components’ Bioavailability and Bioactivity

The bioaccessibility of a compound in food item is defined as the fraction that is released from the matrix of the food in the GI tract to become available for absorption [9]. Saffron has four main bioactive components including crocin (C44H64O24), crocetin (C20H24O4), picrocrocin (C16H26O7) and safranal (C10H14O). The pharmacokinetics of the saffron carotenoids, crocin and crocetin, are known. Their lipophilic character makes them readily absorbable through the intestinal cells, and end up in the chylomicrons without modification before secretion into the bloodstream [10]. In-vitro studies have shown that the crocin is unlikely to be present in the systemic compartment after oral consumption. Indeed, crocin is hydrolyzed rapidly by enzymes in the intestinal epithelium and, to a lesser extent by the intestinal microbiota, leading to deglycosylated trans-crocetin, which is absorbed by the intestinal mucosa [11]. Orally administered crocin leads to up to 81 times higher concentration of crocetin in serum than crocetin oral administration, which is interesting given that the pharmacological effect is often attributed to the trans-crocetin isomer [12]. The in-vivo processing and transformation of crocin to crocetin has major implications. Indeed, many of the in-vitro studies using crocin might not be translatable to in-vivo systems, since crocin, per se, does not reach the bloodstream. Trans-crocetin is the only saffron metabolite that is capable of crossing the blood-brain barrier and reach the CNS (central nervous system), whether pure crocetin or saffron extract is administered [10]. This occurs 90 min after crocetin has been orally administered as reported by Yoshino et al. [13].

## 4. Anti-Inflammatory and Anti-Carcinogenic Effects of Saffron

The anti-inflammatory and anti-oxidant characteristics of crocetin and crocin were evaluated in-vivo in various organs including stomach, intestine, liver and kidney [14,15,16,17,18,19]. Saffron components were shown to down-regulate several pro-inflammatory cytokines’ expression. The effects of crocin and crocetin against oxidative stress include reduction of malondialdehyde level, improving the levels of glutathione and anti-oxidant enzymes such as superoxide dismutase, catalase, and glutathione peroxidase, as well as reducing lipid peroxidation. The anti-cancer activity of saffron components seems to occur indirectly through their anti-oxidant and anti-inflammatory action and directly through their anti-proliferative and pro-apoptotic effects [20].

## 5. Effectiveness of Saffron in Digestive Diseases

Several studies have shown effectiveness of saffron components in various digestive inflammatory disorders, as well as digestive cancer prevention and treatment.

### 5.1. Effects on Gastrointestinal Inflammatory Disorders

An increased body of evidence has demonstrated beneficial effects of carotenoid crocin and crocetin on inflammatory disorders including gastritis and peptic ulcer, irritable bowel syndrome, inflammatory bowel disease, and hepatitis.

#### 5.1.1. Gastritis and Peptic Ulcer

The gastroprotective effect of crocin in ethanol-induced gastric injury was shown by EI-Maraghy et al. [21]. Crocin pretreatment increased gastric juice levels. Mucosal prostaglandin E2 (PGE2), interleukin-6 (IL-6) and TNF-α levels. Myeloperoxidase activity and heat shock protein 70 mRNA and protein levels decreased. The mucosal levels of glutathione, malondialdehyde, and superoxide dismutase activity were restored with crocin treatment. Crocin pretreatment reduces ethanol-induced mucosal apoptosis by down-regulating cytochrome c and caspase-3 expression, decreasing caspase-3 activity and mitigating DNA fragmentation. The study [21] concluded that crocin protects rat gastric mucosa against ethanol-induced injury by displaying anti-inflammatory, anti-oxidative, anti-apoptotic and mucin-secretagogue mechanisms, probably mediated by increased PGE2 release.

With its anti-oxidant properties, saffron has the potential to act as a preventive agent of gastric mucosa damage through enriching glutathione levels and reducing lipid peroxidation [22]. Inoue et al. reported that saffron can inhibit ulcers that are induced by stress and histamines [23]. Similarly, Al-Mofleh also showed that saffron has significant anti-secretory and anti-ulcer activities [24]. These findings need to be further studied through clinical trials to compare saffron to other therapies in peptic ulcer treatment.

#### 5.1.2. Irritable Bowel Syndrome

Gastrointestinal (GI) diseases affect about 60–70 million US citizens on a yearly basis [25]. In 2004, there were an estimated 4.6 million hospitalizations, 72 million ambulatory care visits, and 236,000 deaths attributable to GI diseases [25]. Spending on GI diseases in the US has been estimated at $142 billion per year [25]. Irritable bowel syndrome (IBS) is characterized by abdominal discomfort and bowel habits changes, affecting 30 to 45 million US adults each year [26]. The prevalence of IBS in North America is estimated at 3–20%; however, many studies are reporting numbers as high as 10–15% [27,28]. A prevalence of 11.5% is reported in Western Europe [29]. The peak age for IBS symptoms is between 35–44 years [27]. Two thirds of IBS patients are women [27,30]. Women generally develop symptoms at an earlier age than men (25.6 vs. 30.5 years) [31]. Caucasians are reportedly 2.5 times more likely to develop IBS than African Americans [32]. However, overall prevalence of IBS by ethnic groups in the US has not been reported [33]. It is worth noting that many cases of IBS are more of a psychological nature and are primarily triggered by anxiety and depression [34].

Curcumin, known as Indian saffron has anti-oxidant and anti-inflammatory characteristics and was proved to have beneficial effects in patients with IBS. A recent meta-analysis has highlighted its potential in treating patients with IBS [35]. Its beneficial effects were explained through its anti-oxidant and anti-inflammatory features that are similar to those in saffron as well as to its alteration of the gut microbiota, a point we are discussing below specifically for saffron. Through its anti-depressant properties, saffron would likely be more beneficial to treat IBS patients as it will have the ability to address both components of the IBS in the GI tract as well as those stemming from the nervous system in anxiety and depression scenarios. Considering the potential association of psychological stressors and depression with many cases of IBS, a recent study compared efficacy of saffron extract and fluoxetine (anti-depressive) in improving IBS patients’ quality of life [36]. Results of this study demonstrated no significant difference between saffron and fluoxetine in increasing quality of life as well as decreasing depression and anxiety in this group of patients [36] (Table 1). Therefore, saffron may substitute for the IBS treatment with no toxicity.

Since IBS is associated with a low-grade inflammation of intestinal mucosa, the anti-oxidant and anti-inflammatory effects of saffron components may be useful in improving both inflammation and intestinal symptoms of patients with IBS. It would also be useful to evaluate the effect of saffron on mast cells in the intestinal mucosa of IBS patients. Mast cells increase seems to play an important role in the pathogenesis of symptoms [37]. Another important aspect would be the evaluation of the interaction of saffron components with diet FODMAP (Fermentable Oligosaccharides Disaccharides Monosaccharides and Polyols), also implicated in the symptoms of IBS patients through their action on gut microbiota [38].

#### 5.1.3. Inflammatory Bowel Diseases

Kawabata et al. reported that 4 weeks crocin feeding was able to inhibit Dextran Sulfate Sodium (DSS)-induced colitis and decrease tumor necrosis factor α expression, interleukin- (IL-) 1β, IL-6, interferon γ, NF-κB, cyclooxygenase-2, and inducible nitric oxide synthase in the colorectal mucosa and increased Nuclear factor (erythroid-derived 2)-like 2 (Nrf2) expression [15]. Results suggested that crocin suppressed chemically induced colitis and colitis-associated colon carcinogenesis in mice, at least partly by inhibiting inflammation and the expression of certain pro-inflammatory cytokines and inducible inflammatory enzyme [15]. In a recent study, crocin demonstrated anti-ulcerogenic and coloprotective effects. The therapeutic impact was mediated primarily via enhancement of colon Nrf2 content by 211% in protective protocol and by 350% in curative protocol and HO-1 signaling by 49% in protective protocol and, 288% in the curative protocol. Enhancement of Nrf2 and HO-1 signaling and down-regulation of caspase-3 activity are believed to underly the observed therapeutic effect [16]. Therefore, crocin could be a candidate for both the prevention of colitis and inflammation-associated colon cancer.

#### 5.1.4. Hepatitis

Saffron has shown hepatoprotective effects mainly through its anti-oxidant properties. Rahami et al. reported an effect of ethanol saffron extract (dried stigmas of *Crocus sativus* L.) on hepatic tissue injury in streptozotocin-induced diabetic rats [39]. Saffron ameliorated anti-oxidant enzymes and suppressed lipid peroxidation and nitric oxide formation in aged male rat liver [17]. Ethanol saffron extracts displayed no toxicological effects, and had no significant changes in liver and kidney functions [40]. An in-vivo evaluation by Chen et al. revealed a marked anti-oxidant capacity of crocetin and crocin in liver [18]. Stress-induced oxidative damage of liver was shown to be inhibited by crocin [19]. From the above studies, it appears that the hepatoprotective effects, anti-oxidants, and anti-inflammatory components of saffron could be beneficial in various forms of liver injury such as drug-induced liver damage, viral hepatitis, and alcoholic and non-alcoholic steatohepatitis.

### 5.2. Effects on Cancer Prevention and Treatment

Cancer prevention with natural products represents a promising strategy against cancer development and progression. Natural products with anti-oxidative and anti-inflammatory properties are evaluated for their abilities to block tumor growth and maintain tissues homeostasis [41]. Studies using animal models and human cancer cell lines have demonstrated anti-cancer activities of saffron as well. A summary of these studies [42,43,44,45] and their mode of action on different cancers is shown in Table 2 and Table 3. Investigations have shown that the major carotenoids of saffron extract, crocin, and crocetin arrest cell cycle at S phase, G0/G1and G2/M stage [46,47] inhibiting mitosis, cell proliferation and triggering apoptosis (Figure 1).

It was reported [50] that crocetin inhibited the proliferation of MIA-PaCa-2, BxPC3, Capan-1, and ASCPC-1 cells. Crocetin significantly inhibited cell distribution in S phase impairing DNA replication. This confirms the inhibition of DNA synthesis in crocetin-treated pancreatic cancer cells. The cell cycle entry of cells depends on the activity of several regulatory proteins including Cdc-2, Cdc-25c, cyclin B1 as well as other proteins. In addition, a study demonstrated that anti-cancer activity of saffron can activates the intrinsic and extrinsic routes caspase pathway to lead to the apoptosis of tumor cells [55] (Figure 2).

Apoptosis is a gene regulated process that is important both in normal and pathological conditions. Its regulatory mechanisms include caspases and bcl-2 family proteins [29]. Indeed, saffron extracts induced a p53-dependent pattern with cell cycle arrest at G2/M in HCT116 p53 wild type cells. However, in HCT116 p53−/− cells, it induced a remarkable delay in S/G2 phase transit with entry into mitosis. The apoptotic Pre-G1 cell fraction, Annexin V staining and caspase 3 cleavage showed a more pronounced apoptosis induction in p53+/+ cells. The significantly higher DNA damage, reflected by γH2AX protein levels in p53−/− cells, was coped with by up-regulation of autophagy. Saffron-induced LC3-II protein level was a remarkable indication of the accumulation of autophagosomes, a response to the cellular stress condition of drug treatment [29].

Most of the previously cited studies have used crocin or crocetin, while some used total saffron extracts. This overshadowed the potential roles of some other saffron extracts such as safranal. Amin et al. have recently studied anti-cancer effects of safranal in hepatocellular carcinoma (HCC). Their experiments showed a DNA damage repair and apoptosis unique safranal-mediated cell cycle arrest at G2/M phase and a pronounced effect on DNA damage machinery. Safranal also activated both intrinsic and extrinsic initiator caspases where ER-stress was evidently a major mediator. Their gene set enrichment analysis revealed that unfolded protein response (UPR) is consistently among the top up-regulated genes in the presence of safranal [55]. Crocin can also prevent adverse Cisplatin and Cyclophosphamide effects such as oxidative damage, inflammation and organ toxicity (e.g., hepatotoxicity) [53,56].

All these findings point to the presence of different synergistic anti-cancer, pro-apoptotic and anti-oxidative effects of the different ingredients of saffron of which the cumulative effects are summed up in the anti-oxidative and anti-inflammatory characteristics of total saffron extracts. If specific ingredients of saffron were to be used in the future, ingredient’s specific models of action ought to be determined. However, since saffron is a natural product with no known side effects, its administration as a crude or as total extracts is likely the way to go.

#### 5.2.1. Gastric Cancer

Bathaie et al. [54] reported saffron aqueous extract’s beneficial effects on 1-methyl-3-nitro-1-nitrosoguanidine-induced gastric cancer in rats. Indeed, it inhibited the progression of gastric cancer (20% of cancerous rats treated with higher doses of saffron were completely normal at the end of the experiment and no rat with adenoma in the treated groups was noted). Anti-oxidant, anti-proliferative and apoptotic activities have been reported for crocetin against gastric cancer [54]. In a study by Hoshyar et al. [57], crocin’s mechanism of action was investigated in AGS gastric cancer cells. Crocin proved to be cytotoxic to AGS cells in a dose and time dependent manner. Flow cytometry and caspase activity assessment further confirmed crocin-induced apoptosis. The increased sub-G1 population and stimulated caspases in the treated AGS cancer cells confirmed its anti-cancer effect. He et al. also reported crocetin-induced apoptosis in a different human gastric cancer cell line (BGC-823) [47]. 

#### 5.2.2. Colorectal Cancer

Two saffron anti-cancer activities against colorectal cancer have been reported: anti-proliferative; and pro-apoptotic [42,43,52,58]. Crocin significantly blocked the growth of colorectal cancer cells (HCT-116, SW-480 and HT-29) while not affecting normal cells [43]. Another study has investigated the dependency of saffron’s mechanism of action on p53 in two p53 isogenic HCT116 cell lines and showed induction of DNA damage and apoptosis in both cell lines. However, autophagy has delayed the induction of apoptosis in HCT116 p53 deficient cells [42].

An interesting study [15], showed the potential inhibitory effects of crocin against inflammation-associated mouse colon cancer and chemically induced colitis in male mice. It was found that the development of colonic adenocarcinomas was significantly reduced in azoxymethane mouse model (AMO). There was decreased expression of NF-κB but increased expression of Nrf2 in adenocarcinoma cells. In dextran-sulfate-sodium-induced colitis murine model, dietary crocin significantly decreased the expression of pro-inflammatory cytokines and inducible inflammatory enzymes such as TNF-α, IL-1β, IL-6, IFN-γ, NF-κB, cyclooxygenase-2, and inducible nitric oxide synthase and increased Nrf2 expression. Based on these findings, crocin can be used to prevent colitis and inflammation-associated colon carcinogenesis.

Amerizadeh et al. explored the therapeutic potential of crocin or its combination with 5-flurouracil in a mouse model of colitis-associated colon cancer [59]. Two-dimensional and three-dimensional cell-culture models were used to assess the anti-proliferative and migratory activity of crocin. The results showed that crocin modulates the WNT-pathway and E-cadherin leading to a reduction of cell-growth and invasive behavior of colorectal cancer cells. DSS-induced colonic inflammation was suppressed by crocin as reflected by inflammation score, crypt loss, pathological changes, and histology scores. 

Amin et al. reported that defective autophagosome formation in p53-null colorectal cancer reinforces crocin-induced apoptosis [60]. Rastgoo et al. reported an anti-tumor activity of PEGylated nanoliposomes containing crocin in mice bearing C26 colon carcinoma [61]. It was also reported that saffron extract and its major constituent (crocin) significantly inhibited the growth of colorectal cancer cells while not affecting normal cells [43]. Long term treatment with crocin enhanced survival in rats with colon cancer without major toxic effects [58]. Taken together, all these data indicate that saffron should be considered as a potential cancer preventive agent in clinical trials [62]. It would be useful to evaluate the interaction of its constituents with other components of the diet involved in the carcinogenesis of the colon such as red meat, fiber, vitamins and minerals [63].

#### 5.2.3. Liver Cancer

Saffron was shown to have some effects against HCC both in-vitro and in-vivo [45]. In HCC model, crocin’s anti-inflammatory effect is attributed to its effect on NF-κB signaling pathway. In-vitro analysis confirmed crocin’s effect in HepG2 liver cancer cells by arresting the cell cycle at S and G2/M phases, inducing apoptosis and decreasing inflammation. Network analysis identified NF-κB as a potential major regulatory protein, and as such a potential therapeutic drug target. Crocin inhibited the formation of pre-neoplastic foci of altered hepatocytes in Diethyl nitrosamine-induced HCC models, accompanied with reduced oxidative stress and restored levels of anti-oxidants. Treatment with crocin decreased the activity of some inflammatory markers (COX-2, iNOS, NF-κB, TNF-alpha and its receptor p-TNF-R1) (Figure 3). Crocin exhibited anti-inflammatory properties where NF-κB, among other inflammatory markers, was inhibited [45]. We performed an in-vitro study on T cell effector and B cell antibody responses, along with viability and proliferation capacity, in presence of 0.04%, 2% and 10% saffron. The results showed increases in saffron concentration was met with mild impairment of cell viability. Also, for adaptive immune responses, there was a large increase in T helper type 2 and B cell responses. In parallel, there was a moderate, but not as major, decrease in T effector type 1 polarized responses. Thus, saffron would seem to shift adaptive immune responses towards a more T helper 2-associated B cell response that is associated with bacterial and parasitic (extracellular) pathogen defenses.

#### 5.2.4. Pancreatic Cancer

It has been demonstrated that crocin and crocetin have anti-cancer activity also against pancreatic cancer [49,50]. Bakshi et al. [50] reported that crocin induced cell growth reduction and development of cell apoptosis and G1-phase cell cycle arrest of human pancreatic cancer cell lines (BxPC-3). Dhar et al. [49] demonstrated in a xenograft mouse model that crocetin inhibits pancreatic cancer cell proliferation and tumor progression. They evaluated Cdc-2, Cdc-25C, Cyclin-B1 and EGFR in MIA-PaCa-2 cells, Capan-1, and ASPC-1 pancreatic cancer cells. Crocetin-induced apoptosis occurred through an increase in Bax/Bcl-2 ratio [49].

### 5.3. Effects of Saffron on the Gut Microbiome 

As previously mentioned, a meta-analysis study showed that curcumin (Indian saffron) mode of action is mediated in part through its ability to modulate the gut microbiome [35]. As such, we conducted a pilot study to assess saffron’s effects on the gut microbiome composition in rats. Rats were given saffron in their drinking water (~120 mg/day). Stool samples were collected before and after 4 weeks and stool DNA extracts were used to analyze the gut microbiome structure and changes in the two groups (Control group which drank plain water and case group). Ten stool samples from each of the two groups were used to extract DNA samples for bacterial community analysis. A Polymerase Chain Reaction (PCR) amplification targeting the 16S rRNA gene using universal bacterial primers was performed and the PCR products were sequenced on an Illumina sequencing machine.

Treatment with saffron led to major changes at the phylum level. A dramatic reduction/depletion of *Cyanobacteria and Proteobacteria* and a less dramatic decrease in *Bacteroidetes* and *Firmicutes* phyla within the saffron treated rats was noticed. These reductions were accompanied by an enrichment in *Spirochaetes*, *Tenericutes*, and *Candidatus saccharri* bacteria phyla (Figure 4) in these rats. A linear Discriminant Analysis (LDA) further highlighted the changes observed at the phylum level and pinpointed the specific species within these phyla with major differential values in the treated vs. control rats and vice-versa (Figure 4B,C). A cladogram analysis (Figure 4D) depicts the changes between the two groups with higher level of penetrance and reflect many groups specifically prevalent in untreated rats while other bacterial groups took over as a result of saffron treatment.

Many questions need to be answered regarding the role of saffron and microbiome dysbiosis. For example: How do these major changes in the gut microbiome translate into anti-oxidant and anti-inflammatory contexts for the colon mucosa and beyond? What parts of these gut microbiome changes do have systemic extensions and effects? Which specific component of saffron is responsible for these gut microbiome major shifts? These are questions that will need to be addressed within specific in-vivo models of disease to establish saffron-gut microbiome-host genetics interactions in health and disease within the gastrointestinal tract.

## 6. Conclusion and Perspective of the Review 

In conclusion, saffron shows beneficial effects in many digestive diseases. The above studies show that the prophylactic and therapeutic characteristics of saffron are mainly done through its anti-oxidant and anti-inflammatory effects, inhibition of cell proliferation, induction of apoptosis, and genoprotective properties. However, most of the studies have been performed in animal models and cancer cell lines. Clinical trial studies are needed in order to define the real beneficial effects of saffron components in the prevention, treatment, and mitigation of many digestive diseases in human subjects. These studies will need to consider saffron components specific activities and assess whether these components have some synergistic cumulative effects. The action of these compounds on the gut microbiome dynamics will also need to be explored within specific disease contexts.

## Figures and Tables

**Figure 1 nutrients-11-00943-f001:**
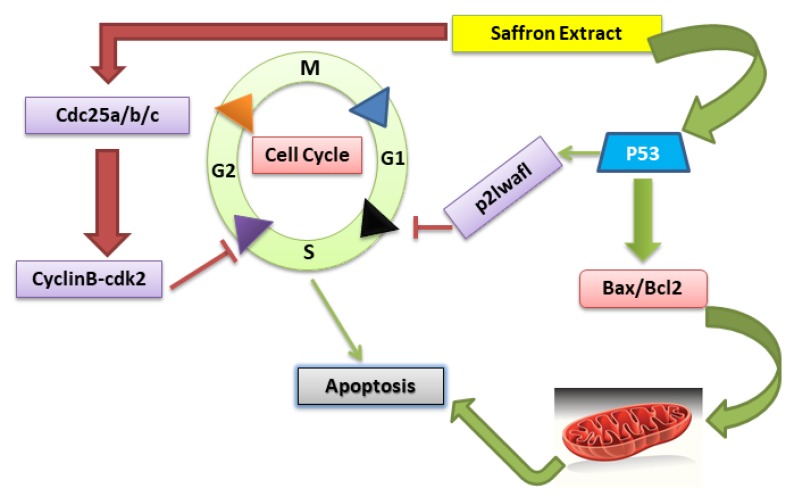
*Crocus sativus* extract affects the cell cycle by blocking the G2 and S phase via p53 and CyclinB-cdk2 proteins. Cancer cells with p53 loss of function have a dysfunctional G1/S checkpoint whereas the G2/M checkpoint may still be functional. When cells are exposed to saffron extract, it causes G1/S arrest via activation of the p53 pathway while G2/M arrest by inhibiting CyclinB−cdk2 (cyclin-dependent kinases), provoking apoptosis of cancer cells. p53 also leads to increasing the Bax and decreasing Bcl2 expression which leads to morphological changes that contributes to apoptosis.

**Figure 2 nutrients-11-00943-f002:**
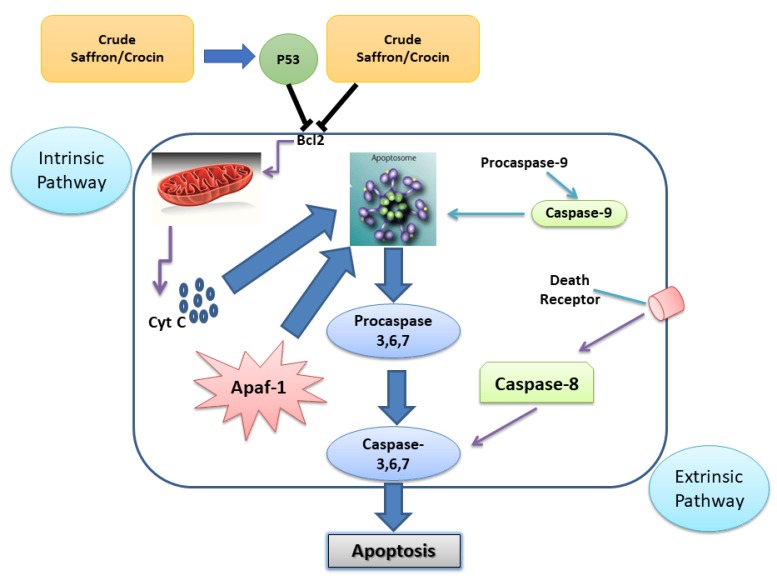
Mitochondrial activation of the intrinsic pathway involves the inhibition Bcl2. Cytochrome C leaking out from the mitochondria to form the complex-apoptosome, composed of caspase-9 and Apaf-1, which activates the executor caspases 3, 6, and 7. Extrinsic route implies the activation of a death receptor in the cytoplasmic membrane by means of a ligand which activates the initiator caspase-8, followed by the executor caspases 3, 6, and 7. The activation of the executor caspases causes morphological changes related to the apoptotic process.

**Figure 3 nutrients-11-00943-f003:**
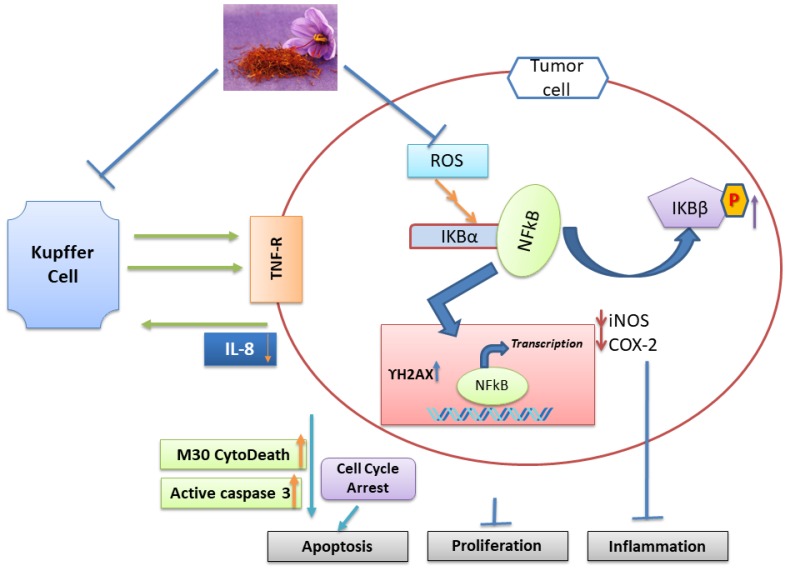
Apoptosis, inflammation, and proliferation alteration by Saffron in liver cells. Both cell lines and animal model data indicate that TNFα receptor alters in Kupper cells. This results in suppression of NF-κB signaling via suppressing the level of iNOS, COX-2, and IL-8. Cell cycle arrest caused by saffron via caspases activation (Figure modified from Amin et al., 2011 [45]).

**Figure 4 nutrients-11-00943-f004:**
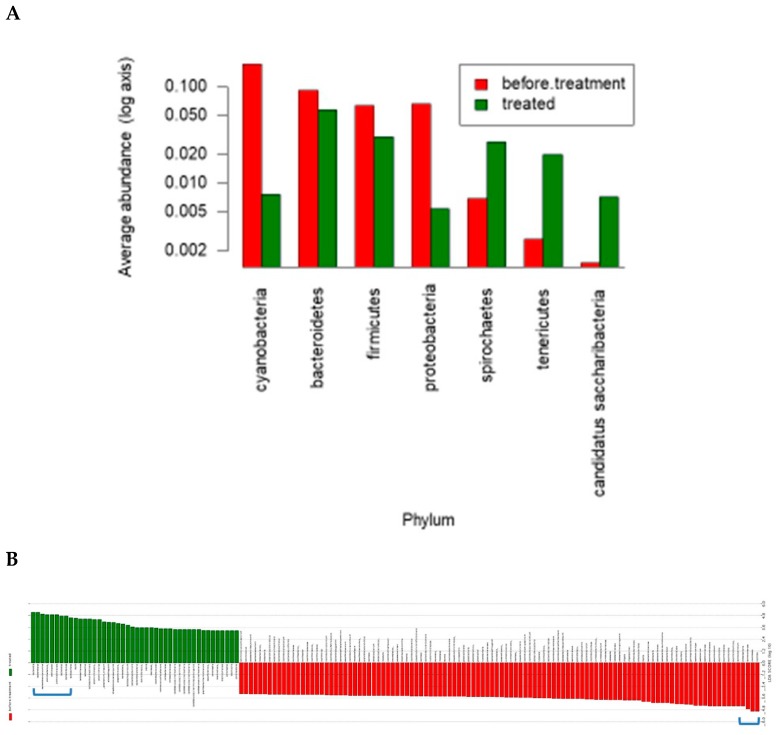
Effect of saffron on the gut microbiome composition. Rats were given saffron in their drinking water (~120 mg/day). Stool samples were collected before and after 4 weeks and stool DNA extracts were used to analyze the gut microbiome structure and changes in the two groups (Control group which drank plain water and case group fed saffron). Ten stool samples from each of the two groups were used to extract DNA samples for bacterial community analysis. A PCR amplification targeting the 16S rRNA gene using universal bacterial primers was performed and the PCR products were sequenced. Treatment with saffron led to major changes at the phylum level. (**A**) Average abundance of different microbiomes before and after saffron treatment in rats. Treatment with saffron led to major changes at the phylum level. Red bars indicate bacterial phyla in untreated while the green bars show the prevalence of different phyla in saffron treated rats. A dramatic reduction of *Cyanobacteria* and *Proteobacteria* and a less dramatic decrease in *Bacteroidetes* and *Firmicutes* phyla within the saffron treated rats was noticed. These reductions were accompanied by an enrichment in *Spirochaetes*, *Tenericutes*, and *Candidatus saccharri* bacteria. (**B**). Linear discrimination analysis (LDA) of Operation Taxonomy Units (OTUs) (Species) indicates major changes in the gut microbiome as a result of saffron treatment: Green bars indicate the discriminant OTUs in untreated while the red bars show the discriminant OTUs in saffron treated rats. (**C**). Magnification of the discriminant OTUs in the rat gut microbiome from (**B**) (brackets). Green bars show that top discriminant OTUs in saffron treated rats while red bars show discriminant OTUs in untreated rats. Top discriminant OTUs in treated rats were from the *Bacteroides* to *Lactobacilli* strains while top discriminant OTUs in untreated rats were from the *Clostridiales* to *Lachnoclostridiale* strains. (**D**). Penetrance of bacterial composition changes as a result of saffron treatment in rats. From Inner to Outer: Phyla to Species. Green or red lines stemming from the center of the cladogram reflect higher penetrance changes as a result of saffron treatment.

**Table 1 nutrients-11-00943-t001:** Primary and secondary outcome measurements for the treatment with Saffron vs. Fluoxetine.

Questionnaire	Weeks into Treatment	Treatment Group					
		Saffron			Fluoxetine		
		Mean	SD	*p*-Value *	Mean	SD	*p*-Value *
IBS-Qol	Baseline	60.00	9.15		59.18	7.28	
	2 weeks	60.61	8.07	0.033	59.27	6.21	0.753
	4 weeks	62.36	7.27	<0.001	61.33	6.74	<0.001
	6 weeks	68.06	7.00	<0.001	67.36	7.58	<0.001
HADS-Depression domain	Baseline	7.48	1.80		7.88	1.85	
	2 weeks	7.36	1.52	0.525	7.45	1.64	<0.001
	4 weeks	6.58	1.23	<0.001	6.76	1.12	<0.001
	6 weeks	5.91	0.98	<0.001	6.21	0.86	<0.001
HADS-Anxiety domain	Baseline	7.27	1.72		7.45	1.60	
	2 weeks	7.03	1.76	0.058	7.61	1.34	0.201
	4 weeks	6.79	1.62	0.001	7.39	1.30	0.690
	6 weeks	6.55	1.50	<0.001	6.94	0.90	0.019

SD = standard deviation, IBS = irritable bowel syndrome, QoL = quality of life, HADS = Hospital Anxiety and Depression Scale * *p*-values are for paired sample t-tests comparing values to their baseline amount.

**Table 2 nutrients-11-00943-t002:** Effects of saffron on liver and colorectal cancer (CRC) cell lines and mechanisms of action.

Types of Cancers	Cell Lines/Animal Model	Mechanism of Action	Reference
Colon Cancer	HCT116	Induction of Apoptosis	[42]
	HCT116, SW480, and HT29	Induction of cytotoxicity and Inhibition of cell proliferation	[43]
Liver Cancer	HPG2	Induction of cytotoxicity and Inhibition of cell proliferation	[44]
	HPG2	Induction of Apoptosis	[45]

**Table 3 nutrients-11-00943-t003:** Molecular mechanisms by which extracts of saffron exert anti-cancer activity in GI cancers.

Type of Cancer	Secondary Metabolite	Mechanism of Action	Molecular Changes	References
Hepatic Cancer	crocin	Apoptosis	Down-regulation of hTERT gene Down-regulation of the expression of catalytic subunit of enzyme telomerase	[44,45,48]
		Anti-oxidant property and Anti-inflammatory effect	Increased the levels of GST, SOD, and CAT Reduced myeloperoxidase activity, malondialdehyde Inhibition of COX 2, iNOS, NF-κB	
Pancreatic Cancer	crocetin	Cell cycle arrest at G2/M Phase	Reduced expression of Cdc-2 (hyperphosphoryltion) Reduced expression of Cdc-25c phosphatase Inhibition of Cyclin B1	[49]
	crocin, crocetin	Apoptosis	Increased expression of Bax protein Suppressed expression of Bcl-2 Elevated Bax/Bcl-2 ratio	[49,50,51]
	crocetin	Inhibition of cell proliferation	Reduced activity of EGFR Reduced phosphorylation of Akt	[49,51]
Colorectal Cancer	crocetin	Cell cycle arrest at S Phase	Reduced expression of cyclin A and cdk2	[52]
	crocin	Cell cycle arrest at G3 phase	Decrease in the levels of cyclin B1 and pH3	[42,45]
	Crocin, crocetin	Apoptosis	Augmented expression of p53 and P21	[42,53]
	crocin	DNA Damage	Up-regulation of H2AX	[42,45]
		Autophagolysis	Formation of LC3-II Decrease in protein levels of Beclin 1 and Atg 7 genes	[45]
Gastric Cancer	crocin	Apoptosis	Activation of caspases Elevated Bax/Bcl-2 ratio	[54]

GST: Glutathione S-transferases, SOD: Superoxide dismutase, CAT: Catalase, EGFR: Epidermal growth factor receptor.

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
