# Peer review of "Saffron: The Golden Spice with Therapeutic Properties on Digestive Diseases"

_nutrients, 2019, doi:10.3390/nu11050943_

Reviewer 1 Report

The review article “Saffron: the Golden Spice with Therapeutic Properties on Digestive Diseases” by Ashktorab et al hardly provides any newer information to the scientific community. In this article, the authors tried to focus on the mechanisms of action and the medical use of saffron, particularly in digestive diseases. The manuscript is well written and well organized. But, there are ample amount of review articles available and are focusing similarly on this subject area (for example: Therapeutic effects of saffron (Crocus sativus L.) in digestive disorders: a review by Khorasany et al, Iran J Basic Med Sci. 2016 May; 19(5): 455–469). Therefore, authors need to highlight and mention the importance of the present study and why their study is unique in comparison to other studies.

Comments:

1. Figure 2 is not well illustrated and needs to incorporate clearer figure legend.

2. Figure legend of “Figure 3” has been copied from “Reference 11” of the manuscript. Authors should rewrite the figure legend.

3. Authors need to incorporate the illustration covering chemical structure of different active ingredients of saffron.

4. Authors should include a “perspective of review” section in their review article.

5.  Authors need to replace Figure 4 with legible and high resolution figure.

6. Typos errors are needed to be checked.

The information packed in this review article are not qualitatively sufficient to be published in this present format. 

Author Response

Reviewer#1:

Comments:

The review article “Saffron: the Golden Spice with Therapeutic Properties on Digestive Diseases” by Ashktorab et al hardly provides any newer information to the scientific community. In this article, the authors tried to focus on the mechanisms of action and the medical use of saffron, particularly in digestive diseases. The manuscript is well written and well organized. But, there are ample amount of review articles available and are focusing similarly on this subject area (for example: Therapeutic effects of saffron (Crocus sativus L.) in digestive disorders: a review by Khorasany et al, Iran J Basic Med Sci. 2016 May; 19(5): 455–469). Therefore, authors need to highlight and mention the importance of the present study and why their study is unique in comparison to other studies.

 Response: The paper by Khorasani et al. has put together the effect of saffron all over the body (table 1). Their approaches to GI diseases were different in 2015. Our review paper included the data from microbiome from our laboratory and liver cancer part. All the investigators in our review are well known GI clinical and translational experts which give the paper a higher standard for the impact of the saffron on the outcome, both in prevention and interventional approaches.

1. Figure 2 is not well illustrated and needs to incorporate clearer figure legend.

Response: Figure 2 legend has been revised.

2. Figure legend of “Figure 3” has been copied from “Reference 11” of the manuscript. Authors should rewrite the figure legend.

Response: Figure 3 legend has been revised as below:

Figure 3. Apoptosis, inflammation, and proliferation alteration by Saffron in liver cells. Both cell lines and animal model data indicates that TNFα receptor alters in Kupper cells. This results in suppression of NF-kB signaling via suppressing the level of iNOS, COX-2, and IL-8. Cell cycle arrest caused by saffron via caspases activation (Figure modified from Amin et al., 2011, Ref#43).

3. Authors need to incorporate the illustration covering chemical structure of different active ingredients of saffron.

Response: We have added the following in the introduction for the main bioactive components of saffron. Saffron has four main bioactive components including crocin (C44H64O24), crocetin (C20H24O4), picrocrocin (C16H26O7) and safranal (C10H14O).

4. Authors should include a “perspective of review” section in their review article.

Response: A new section has been added as suggested by the reviewer.

Comments:

5.  Authors need to replace Figure 4 with legible and high-resolution figure.

Response: Figure 4 is in high resolution in the current format. However, we made each part as an individual figure for clearer presentation.

6. Typos errors are needed to be checked.

Response: The manuscript has been reviewed and typos corrected.

Reviewer 2 Report

This review by Ashktorab et al. reviewed the molecular effects of saffron in variety of digestive diseases. This review manuscript is well-written, concise and informative. Figures are helpful in grasping the mechanisms. The reviewed studies used animal models and cell culture, calling for human trials. I have no major comments. A few minor points that authors could consider to improve the MS:

-Figures: There might be an error the  pro-apoptotic protein (p53), activating an anti-apoptotic protein (Bcl2) which leads to apoptosis.

Similarly in the figure 2, authors aim to show apoptotic activity via caspase pathway activation by P53. However the figure gives the impression that P53 is inhibiting the caspase.

The arrows’ flow in these figure better to be reviewed again by the authors and corrected as needed.

-This article would be a good place to mention possible side effects of saffron in the published studies. If no side effects in any of the reviewed studies is noted, this can be stated in the manuscript. 

Author Response

-Figures: There might be an error the  pro-apoptotic protein (p53), activating an anti-apoptotic protein (Bcl2) which leads to apoptosis. Similarly in the figure 2, authors aim to show apoptotic activity via caspase pathway activation by P53. However, the figure gives the impression that P53 is inhibiting the caspase. The arrows’ flow in these figure better to be reviewed again by the authors and corrected as needed.

Response: The figures legend has been revised to address the reviewer’s comments regarding Bax/Bcl2. We mentioned the Bax/Bcl2 ratio increase which refers to increasing of Bax and decrease expression of Bcl2.

Figure legend 1 was revised as below: When cells are exposed to saffron extract, it causes G1/S arrest via activation of the p53 pathway while G2/M arrest by inhibiting CyclinB−cdk2 (cyclin-dependent kinases), provoking apoptosis of cancer cells. P53 also leads to increasing the Bax and decreasing of Bcl2 expression which leads to morphological changes that contribute to apoptosis.

Figure 2 is correctly referring to the inhibitory action of Bcl2, which in turn act on the release of cytochrome C from mitochondria. Cytochrome-C leaking out from the mitochondria to form the complex − apoptosome, composed of caspase-9 and Apaf-1, which in turn activates the executor caspases − 3, −6, and −7. Therefore, we do not think any changes necessary.

Reviewer 3 Report

Manuscript must be revise

Nutrients -466716
The present manuscript is the review of Saffron: the Golden Spice with Therapeutic Properties on Digestive Diseases
The manuscript contains is consistent but need corrections and must be improves.
For example all the must be improves by increasing the quality (pixels)

1). Methods
The number of keywords used was no sufficient. The authors can add the following keywords: dyspepsia, constipation, diarrhoea, colic, Stomach ulcer, gastritis, hepatitis

3. Saffron Components Bioavailability and Bioactivity:
Please check the text.
Line 1 : Compound jn food
Line 9 : to81

4. Anti-Inflammatory and Anti-Carcinogenic Effects of Saffron
In all the text, pleased replace in-vivo by in vivo and Crocus sativus in italic characters

5.2. Effects on Cancer Prevention and Treatment
Figure 1. Effect of Crocus sativus extract in the cell cycle by blocking the G1 not the G2

5.2.2. Colorectal Cancer: In dextran‐sulfate‐sodium induced induced colitis

All the must revised carefully

Author Response

Comments:

The present manuscript is the review of Saffron: The Golden Spice with Therapeutic Properties on Digestive Diseases. The manuscript contains is consistent findings but needs corrections and must be improved. For example all the must be improves by increasing the quality (pixels) of the figures.

Response: Figures have been revised and made clearer with high resolution.

1. The number of keywords used was no sufficient. The authors can add the following keywords: dyspepsia, constipation, diarrhea, colic, Stomach ulcer, gastritis, hepatitis

Response: We thank the reviewer for the suggestion. New keywords have been added.

2. Saffron Components Bioavailability and Bioactivity:

Response: We added the bioavailability component in the introduction.

-Line 1 : Compound jn food

-Line 9 : to81

Response: We corrected the typos.

4. Anti-Inflammatory and Anti-Carcinogenic Effects of Saffron

In all the text, pleased replace in-vivo by in vivo and Crocus sativus in italic characters.

Response: We replaced in-vivo with in vivo and italicized Crocus sativus.

5.2. Effects on Cancer Prevention and Treatment

Figure 1. Effect of Crocus sativus extract in the cell cycle by blocking the G1 not the G2

Response: We have correctly indicated and explained in the text and based on our own studies (Amr et al) that G2 phase of cell cycle affected through CyclinB−cdk2 (cyclin-dependent kinases), provoking apoptosis of cancer cells.

5.2.2. Colorectal Cancer: In dextran‐sulfate‐sodium induced colitis

Response: The manuscript carefully reviewed and has been revised based on your positive comments.

Round  2

Reviewer 1 Report

The review article “Saffron: the Golden Spice with Therapeutic Properties on Digestive Diseases” by Ashktorab et al have been revised and addressed all the comments.

1. Figure 4B and 4D are still not legible. Authors are requested to replace these with better versions.